# ANALYSIS OF ALIGNMENT PHENOMENON IN SIMPLE TEACHER-STUDENT NETWORKS WITH FINITE WIDTH

## ABSTRACT

Recent theoretical analysis suggests that ultra-wide neural networks always converge to global minima near the initialization under first order methods. However, the convergence property of neural networks with finite width could be very different. The simplest experiment with two-layer teacher-student networks shows that the input weights of student neurons eventually align with one of the teacher neurons. This suggests a distinct convergence nature for "not-too-wide" neural networks that there might not be any local minima near the initialization. As the theoretical justification, we prove that under the most basic settings, all student neurons must align with the teacher neuron at any local minima. The methodology is extendable to more general cases, where the proof can be reduced to analyzing the properties of a special class of functions that we call *Angular Distance (AD) function*. Finally, we demonstrate that these properties can be easily verified numerically.

## 1 INTRODUCTION

The theoretical understanding to the training of neural networks has been a major and long-standing challenge. A recent line of research (Du et al., 2019c;b;a; Arora et al., 2019a;b) presented a major theoretical breakthrough via an elegant approach to characterize the training procedure of ultra-wide neural networks. At the high level, the training loss profile of an ultra-wide neural network ($m = \Omega(N^6)$, $N$ is the size of the training set) uniformly converges to zero. During the training procedure, almost all neurons' weight vectors remain close to their initialization, which in term preserves the uniform converging rate. As a result, the training loss converges to zero with almost all neurons' weight vectors near their initialization.

Does the elegant theory reveal the fundamental mechanics behind the success of practical neural networks whose widths are finite ($m = O(N)$)? The theory suggests a clear property that is critical to the uniform convergence: *The weight vectors of the neurons rarely move away from their initialization.*

In this paper, we first examine this conjecture with experiments. Unfortunately, a simple experiment with two-layer teacher-student networks exhibits contradicting properties: *Despite of their randomized initialization, the weight vectors of the student neurons eventually align with the weight vector of one of the teacher neurons.*

In other words, almost all the student weight vectors end up somewhere far away from where they begin: from the randomized initialization to some specific teacher weight vector. We emphasize that such experiments with teacher-student models are generic in the sense that, according to the universal approximation theorem, any target function can be equivalently described as a two-layer neural network with ReLU activation. Although such a teacher network may not be accessible for an arbitrary dataset, the teacher-student model is sufficiently representative for empirical justification.

Similar alignments are also observed for over-parameterized student and teacher networks of multiple layers (Tian, 2020). In other words, the neuron weights alignment might be the more appropriate fundamental mechanics for neural networks with finite width. In fact, once the student-teacher alignment has been established, it is straightforward that the training loss converges to zero.

We then investigate the other direction of the observation: *Does the convergence of gradient descent in this case imply a perfect alignment between the student and the teacher neurons?* To the best

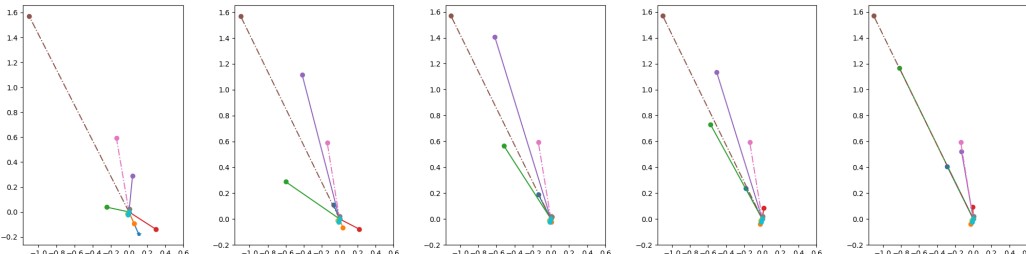

Figure 1: The weight vectors of teacher neurons (dotted lines) and student neurons (solid lines) in epoch 0, 10, 1000, 5000, 50000 respectively.

of our knowledge, the theoretical understanding to this question is extremely limited even for the simplest 1-teacher 2-student case.

In this paper, we initiate the study of the fundamental yet highly non-trivial question with the most basic 1-teacher $m$-student setting. Both the teacher and students are two-layer neural networks with ReLU activation. Mathematically, the question is equivalent to whether there exist any non-alignment solutions to the set of equations defining zero gradients.

We demonstrate a complete proof of non-existence for $m = 2$, and rule out the existence of solutions with special form for general $m \geq 3$: (i) all student neuron weight vectors fall in the same half-plane; or (ii) the angles between any two weight vectors are rational multiples of $\pi$.

For general cases, we show that the theoretical problem can be reduced to the analysis of a special class of functions which we call *angular-distance (AD)* functions: If one of the associated matrices of the AD function has a non-positive determinant, then there is no non-trivial solution to the set of equations. Hence no local minima without perfect alignment.

In light of the reduction, we numerically examined the property that the minimum determinant of the associated matrices are always non-positive unless perfect alignment or problem degenerated. Moreover, the minimum determinant behaves as a potential function in the sense that the further from being degenerated, the further from being non-negative.

## 1.1 RELATED WORKS

Recent papers have made tremendous progress on the theoretical analysis of sufficiently over-parametrized neural networks or those with even infinite width. With some mild assumptions, it has been proved that the training error can uniformly converge to zero via gradient descent (Allen-Zhu et al., 2019b; Du et al., 2018b; 2019c;b; Du and Hu, 2019; Lee et al., 2019; Li and Liang, 2018; Zou et al., 2018). Surprisingly, such a uniform convergence implies there always exist global minima near a random initialization, generalization bounds also emerge under the same frameworks (Allen-Zhu et al., 2019a; Arora et al., 2019a; Cao and Gu, 2019). Moreover, as the network width approaches to infinity, fundamental connections between over-parameterized neural networks and kernel methods are also discovered, which hint potential correspondence between the generality of deep learning and that of kernel methods (Arora et al., 2019b; Daniely et al., 2016; Daniely, 2017; Du et al., 2019a; Hazan and Jaakkola, 2015; Jacot et al., 2018; Mairal et al., 2014; Yang, 2019).

Despite the beauty of the theoretical results, empirical studies exhibit certain mismatches with the theory. Nagarajan and Kolter (2019) shows that the uniform convergence property may be insufficient to explain the generalization ability of neural network. Moreover, Tian (2020) considers using teacher networks to train over-parametrized student networks of the same depth and demonstrates that almost all student neurons either align with some teacher neuron or have no contribution to the final output. Our theoretical results derive from analyzing the behaviour of gradient descent assuming that the input distribution is Gaussian, following the methodology commonly used in many recent works (Brutzkus and Globerson, 2017; Du et al., 2017; Du and Lee, 2018; Du et al., 2018a;c; Li and Yuan, 2017; Tian, 2017; Zhong et al., 2017).

## 2 PRELIMINARIES

Consider training a two-layer student network with a two-layer teacher network, where the teacher network has only one neuron in its hidden layer while the student network has $m$ neurons in its hidden layer. Both networks have ReLU activation and we fix their weights of output layers as $1$ to simplify the notations. Let $\boldsymbol{w}_0 \in \mathbb{R}^d$ denote the weights of the teacher's hidden neuron, and $\boldsymbol{w}_1, \ldots, \boldsymbol{w}_m \in \mathbb{R}^d$ denote the weights of each student hidden neuron, respectively. Formally, the output of the teacher and the student network are

$$F_{\text{teacher}}(x, W) = \sigma(\boldsymbol{w}_0^\mathsf{T}\boldsymbol{x}), \quad F_{\text{student}}(x, W) = \sigma(\boldsymbol{w}_1^\mathsf{T}\boldsymbol{x}) + \cdots + \sigma(\boldsymbol{w}_m^\mathsf{T}\boldsymbol{x}).$$

Let $\theta_{ij} \in [0, \pi]$ be the angle between $\boldsymbol{w}_i$ and $\boldsymbol{w}_j$, i.e.,

$$\theta_{ij} = \arccos \frac{\boldsymbol{w}_i \cdot \boldsymbol{w}_j}{\|\boldsymbol{w}_i\|_2 \|\boldsymbol{w}_j\|_2},$$

$\boldsymbol{u}_i = \boldsymbol{w}_i / \|\boldsymbol{w}_i\|_2$ be the unit-length vector with the same direction as $\boldsymbol{w}_i$, and $\rho_i = \|\boldsymbol{w}_i\|_2$ be the length of $\boldsymbol{w}_i$. In particular, for the 2-dimensional cases, i.e., $d = 2$, we use $\theta_i \in [0, 2\pi)$ to denote the polar angle of $\boldsymbol{w}_i$, and $\boldsymbol{u}(\theta)$ to denote the unit-length vector with polar angle $\theta$.

A student neuron $\boldsymbol{w}_i$ is called *aligned* with the teacher, if $\theta_{0i} = 0$, or *negatively aligned* with the teacher, if $\theta_{0i} = \pi$. For the 2-dimensional cases, i.e., $d = 2$, being aligned is equivalent to $\theta_i = \theta_0$.

Assume the data follows the normal distribution, i.e., $\boldsymbol{x} \sim \mathcal{N}(\boldsymbol{0}, I)$, then the population loss is

$$l = \mathbb{E}_{\boldsymbol{x} \sim \mathcal{N}(\boldsymbol{0}, I)} \left[ \tfrac{1}{2} \left( \sigma(\boldsymbol{w}_0^\mathsf{T}\boldsymbol{x}) - \sum_{i=1}^{m} \sigma(\boldsymbol{w}_i^\mathsf{T}\boldsymbol{x}) \right)^2 \right].$$

The partial derivative of the squared loss with respect to each student neuron $\boldsymbol{w}_i$ is (Brutzkus and Globerson (2017))

$$\frac{\partial l}{\partial \boldsymbol{w}_i} = \sum_{j=1}^{m} \left( \frac{\pi - \theta_{ij}}{4\pi} \boldsymbol{w}_j + \frac{\rho_j}{4\pi} \sin\theta_{ij} \boldsymbol{u}_i \right) - \left( \frac{\pi - \theta_{i0}}{4\pi} \boldsymbol{w}_0 + \frac{\rho_0}{4\pi} \sin\theta_{i0} \boldsymbol{u}_i \right). \tag{1}$$

In other words, the training process terminates, i.e., the gradient is zero, if and only if

$$(\pi - \theta_{i0})\boldsymbol{w}_0 = \left( \pi\rho_i - \rho_0 \sin\theta_{i0} + \sum_{j \neq i} \rho_j \sin\theta_{ij} \right) \boldsymbol{u}_i + \sum_{j \neq i} (\pi - \theta_{ij})\boldsymbol{w}_j, \forall i. \tag{2}$$

**Observation 2.1.** *If the following two conditions are met, then equation 2 are satisfied:*

    *1. all student neurons ($\rho_i > 0$) are aligned with the teacher,*

    *2. and the sum of their lengths equals to $\rho_0$.*

In this paper, we argue that these two conditions are also necessary for equation 2 to stand. In the remaining part of this paper, we always assume $d = 2$.

## 3 ALIGNMENT PHENOMENON AND THE BASIC CASE

As mentioned, simple experiments show that the student neurons eventually align with teacher neurons under gradient descent. Figure 1 illustrates an example in which the teacher network has 5 neurons. While training via gradient descent with data generated from $\mathcal{N}(\boldsymbol{0}, I)$, the randomly initialized student neurons gradually get aligned with one of the teacher neurons. We call it the *alignment phenomenon*.

In this section, we prove for the most basic case with $m \leq 2$.

### 3.1 BASIC CASE WITH $m \leq 2$

Note that when $m = 1$, the only solution to equation 2 is that the student perfectly aligns with the teacher, i.e., $\boldsymbol{w}_1 = \boldsymbol{w}_0$. Because $(\pi - \theta_{10})\boldsymbol{w}_0 = (\pi\rho_1 - \rho_0 \sin\theta_{10})\boldsymbol{u}_1$ implies that $\boldsymbol{w}_0$ and $\boldsymbol{u}_1$ point towards the same direction. Thus, $\theta_{10} = 0$, which means that $\pi\boldsymbol{w}_0 = \pi\rho_1\boldsymbol{u}_1 = \pi\boldsymbol{w}_1$.

However, the proof of the following theorem for $m = 2$ becomes quite non-trivial.

**Theorem 3.1.** *When $m = 2$, there is no other solution to equation 2 except the trivial one, i.e., $\theta_{01} = \theta_{02} = 0$ and $\rho_1 + \rho_2 = \rho_0$.*

*Therefore, once the training converged, the student neuron(s) must be aligned with the teacher neuron.*

Due to the space limit, we send the complete proof to Appendix A, while highlight the key steps here.

*Proof sketch.* Suppose there exists a solution with $\theta_{12} \neq 0$. In this case, the two equations in equation 2 imply two decomposition of $\boldsymbol{w}_0$ onto $\boldsymbol{u}_1$ and $\boldsymbol{u}_2$. Because such a decomposition is unique for $m = 2 = d$, we know the coefficients of $\boldsymbol{w}_0$, $\boldsymbol{u}_1$, $\boldsymbol{u}_2$ must be proportional. From there, we have

$$\frac{\boldsymbol{w}_0}{\pi - \theta_{12}} = \frac{\boldsymbol{w}_1}{\pi - \theta_{20}} + \frac{\boldsymbol{w}_2}{\pi - \theta_{10}}.$$

By projecting this equation onto $\boldsymbol{w}_0$ and the orthogonal direction of $\boldsymbol{w}_0$, we can further conclude that

$$\frac{\rho_0}{(\pi - \theta_{12})\sin\theta_{12}} = \frac{\rho_1}{(\pi - \theta_{20})\sin\theta_{20}} = \frac{\rho_2}{(\pi - \theta_{10})\sin\theta_{10}}.$$

By plugging the above equation into equation 2, we can reach a contradiction and complete the proof. $\square$

Note that a major challenge of extending this proof to $m \geq 3$ is that the decomposition is not unique anymore, which breaks the first step of the proof.

Even though, it is still useful to notice that whenever two student neurons get aligned, i.e., $\theta_{ij} = 0$, they can be replaced by a single neuron $i^*$ of the same direction and unified length, $\rho_{i^*} = \rho_i + \rho_j$, so that any such a non-trivial solution to $m = k$ implies a non-trivial solution to $m = k - 1$.

Therefore, for general $m \geq 3$, we can without loss of generality prove that there is no solution to equation 2 with $0 = \theta_0 < \theta_1 < \cdots < \theta_m < 2\pi$ and $\rho_0 = 1, \rho_1, \ldots, \rho_m > 0$.

## 4 SPECIAL CASES

In this section, we prove that for $m \geq 3$, there is no solution to equation 2 satisfying either of the two special conditions:

- *Students in a half-plane:* $0 = \theta_0 < \theta_1 < \cdots < \theta_m < \pi$;
- *Polar angles being rational multiples of $\pi$:* $\forall i \in \{1, \ldots, m\}, \theta_i = q_i\pi$ for some rational number $q_i$.

### 4.1 SPECIAL CASE WHEN ALL NEURONS ARE IN THE UPPER HALF-PLANE

We first show the following theorem.

**Theorem 4.1.** *When all vectors of student neurons are in the upper half-plane, i.e., $0 = \theta_0 < \theta_1 < \cdots < \theta_m < \pi$, then equation system equation 2 has no non-trivial solution.*

The rigorous proof of Theorem 4.1 is provided in Appendix B.3. Here, we present two lemmas of function $f(\theta) = (\pi - \theta)\cos\theta + \sin\theta$, and the proof of the following two lemmas can be found in Appendix B.

**Lemma 4.2.** *(decreasing lemma) $f(\theta) = \sin\theta + (\pi - \theta)\cos\theta$ decreases in the interval $[0, \pi]$, and $f(0) = \pi$, $f(\pi) = 0$.*

**Lemma 4.3.** *(log-concave lemma)* $\forall \alpha, \beta \in (0, \pi)$, *and* $0 < \alpha + \beta < \pi$, *it holds that* $f(\alpha)f(\beta) > f(\alpha + \beta)f(0)$.

The above two lemmas reveal some properties of function $f(\theta) = \sin \theta + (\pi - \theta) \cos \theta$. In Section 5, we will show another amazing property of $f$ which correlates to positive semi-definite matrices, and $f$ also applies to analyze the phenomenon behaviour of our neural network in general case.

### 4.2 SPECIAL CASE WHEN ALL THE POLAR ANGLES ARE RATIONAL MULTIPLES OF $\pi$

In this subsection, we consider another special case, and prove that under this constraint, equation system equation 2 has no non-trivial solution.

**Theorem 4.4.** *When all polar angles are rational multiples of $\pi$, equation system equation 2 has no non-trivial solution.*

The proof of Theorem 4.4 is shown in Appendix C. The key point of the proof is the following lemma.

**Lemma 4.5.** *When* $0 < \theta_1 < \theta_2 < \cdots < \theta_m < 2\pi$, *and there does not exist* $i, j$, *s.t.* $\theta_i + \pi = \theta_j$, *then the following* $m \times m$ *matrix* $[\pi - \theta_{ij}]_{i,j=1}^m$ *is positive-definite.*

This lemma directly derives from Theorem 3.1 in Du et al. (2019c). If we define $f(\theta) = \pi - \theta$, then the associated function $g$ is a PAD function (see Definition 5.7), which will be further discussed in Section 5.

Moreover, in this subsection, we assume that all the polar angles are rational multiples of $\pi$. Since rational numbers are dense, equation system equation 2 has no non-trivial solution 'almost everywhere', which indicates that equation system equation 2 has no non-trivial solution without any constraints.

## 5 ANALYSIS OF GENERAL CASE

The general case when $m \geq 3$ without any special constraints is rather difficult to prove. However, if determinant condition (see Theorem 5.1) always holds, then we can rigorously prove that the alignment phenomenon always happens for arbitrary number of student neurons.

Moreover, the determinant condition can be generalized to a novel class of functions – Angular Distance (AD) function (see Subsection 5.2). Since the proof of two special cases in Section 4 are with the aid of AD functions directly or indirectly, and AD function actually appears in the previous literature (e.g., Du et al. (2019c)) although implicitly, it is great likely that AD function can be applied widely to theoretical analysis of neural networks.

### 5.1 DETERMINANT CONDITION

The following theorem clarifies the relationship between determinant condition and alignment phenomenon.

**Theorem 5.1.** *Define* $f(\theta) = (\pi - \theta) \cos \theta + \sin \theta$. *Let* $\boldsymbol{f}_i = [f(\theta_{1i}), f(\theta_{2i}), \cdots, f(\theta_{mi})]^\mathsf{T}$ *for* $0 \leq i \leq m$. *Also, define matrix* $\boldsymbol{M}_i = [\boldsymbol{f}_1, \cdots, \boldsymbol{f}_{i-1}, \boldsymbol{f}_0, \boldsymbol{f}_{i+1}, \cdots, \boldsymbol{f}_m]$ *and* $\boldsymbol{M} = [\boldsymbol{f}_1, \boldsymbol{f}_2, \cdots, \boldsymbol{f}_m]$. *If the following condition (determinant condition) :*

$$\forall m \geq 3, 0 = \theta_0 < \theta_1 < \cdots < \theta_m < 2\pi \text{ with no } 0 \leq i < j \leq m \text{ s.t. } \theta_i + \pi = \theta_j,$$
$$\text{it always holds that } \det \boldsymbol{M} > 0 \text{ and there exists } 1 \leq i \leq m, \text{ s.t. } \det \boldsymbol{M}_i \leq 0,$$

*holds, then equation system equation 2 has no non-trivial solution, i.e., the student neural network converges only when aligning.*

*Proof.* We only consider the general case when $m \geq 3$, since the basic case when $m \leq 2$ has been proved in Subsection 3.1.

If we project $i$-th equation in equation systems equation 2 to the direction $\boldsymbol{u}(\theta_i)$, we will obtain that

$$(\pi - \theta_{i0}) \cos \theta_{i0} \rho_0 = \pi \rho_i - \rho_0 \sin \theta_{i0} + \sum_{j \neq i} \rho_j \sin \theta_{ij} + \sum_{j \neq i} (\pi - \theta_{ij}) \cos \theta_{ij} \rho_j, \forall i,$$

which is equivalent to

$$(\sin\theta_{i0} + (\pi - \theta_{i0})\cos\theta_{i0})\rho_0 = \pi\rho_i + \sum_{j\neq i}(\sin\theta_{ij} + (\pi - \theta_{ij})\cos\theta_{ij})\rho_j, \forall i. \tag{3}$$

Note that $f(\theta) = \sin\theta + (\pi - \theta)\cos\theta$, so equation 3 is equivalent to

$$f(\theta_{i0})\rho_0 = \sum_{j=1}^{m} f(\theta_{ij})\rho_j, \forall i. \tag{4}$$

We can express equation 4 in a matrix form

$$M\rho = b, \tag{5}$$

where

$$M = \begin{bmatrix} f(\theta_{11}) & f(\theta_{12}) & \cdots & f(\theta_{1m}) \\ f(\theta_{21}) & f(\theta_{22}) & \cdots & f(\theta_{2m}) \\ \vdots & \vdots & \ddots & \vdots \\ f(\theta_{m1}) & f(\theta_{m2}) & \cdots & f(\theta_{mm}) \end{bmatrix}, \quad \rho = \begin{bmatrix} \rho_1 \\ \rho_2 \\ \vdots \\ \rho_m \end{bmatrix}, \quad b = \rho_0 \begin{bmatrix} f(\theta_{10}) \\ f(\theta_{20}) \\ \vdots \\ f(\theta_{m0}) \end{bmatrix}.$$

If $\det M > 0$, then by Cramer's rule, $\rho_i = \frac{\det M_i}{\det M}$. And since there exists $1 \leq i \leq m$, s.t. $\det M_i \leq 0$, thus $\rho_i \leq 0$. □

Determinant condition has elegant and simple form, and can be easily verified numerically (see Section 6). Moreover, the analysis of function $f(\theta) = \sin\theta + (\pi - \theta)\cos\theta$ and determinant condition, can be generalized to a novel class of function – Angular Distance (AD) function, which will be further discussed in Subsection 5.2.

## 5.2 AD FUNCTION

In this subsection, we provide the formal definition of angular distance (AD) function, and analyze the relationship between its proposition and the determinant condition in Theorem 5.1.

**Definition 5.2.** *(AD function) Let $f : [0, \pi] \to \mathbb{R}$ be a continuous function, and let $g : [0, 2\pi] \times [0, 2\pi] \to \mathbb{R}$, where $g(x, y) = f(\min\{|x - y|, 2\pi - |x - y|\})$. Then function $g$ is an angular distance(AD) function if $\forall h : [0, 2\pi] \to \mathbb{R}$ that is continuous and real-valued in $[0, 2\pi]$, with $h(0) = h(2\pi)$, it holds that*

$$\int_{x=0}^{2\pi} \int_{y=0}^{2\pi} h(x)g(x,y)h(y)\mathrm{d}x\mathrm{d}y \geq 0.$$

*Also, we call $g$ the associated function of $f$.*

Another important concept is the associated matrices of an AD function.

**Definition 5.3.** *(associated Matrix) Let $g$ be an AD function and give $(m + 1)$ fixed angles $0 = \theta_0 < \theta_1 < \theta_2 < \cdots < \theta_m < 2\pi$. Let $g_i = [g(\theta_1, \theta_i), g(\theta_2, \theta_i), \cdots, g(\theta_m, \theta_i)]^\mathsf{T}$ for $0 \leq i \leq m$. Also, define matrix $M_i = [g_1, \cdots, g_{i-1}, g_0, g_{i+1}, \cdots, g_m]$ and $M = [g_1, g_2, \cdots, g_m]$. We call $M, M_1, \cdots, M_m$ associated matrices of $g$ and given $\theta_0, \theta_1, \cdots, \theta_m$.*

Along with the above two definitions, we provide a proposition and a conjecture of AD functions.

**Proposition 5.4.** *(positive semi-definite) Given $0 = \theta_0 < \theta_1 < \cdots < \theta_m < 2\pi$, for an AD function $g$, its associated matrix $M$ is positive semi-definite.*

**Conjecture 5.5.** *(non-positive determinant) For $m \geq 3$, given $0 = \theta_0 < \theta_1 < \cdots < \theta_m < 2\pi$, for an AD function $g$, and its associated matrices $M_1, M_2, \cdots, M_m$, then there exists $i$, where $1 \leq i \leq m$, s.t. $\det M_i \leq 0$.*

The proof of Proposition 5.4 is shown in Appendix D.1. For Conjecture 5.5, although we are not able to prove it, some experimental results (see Section 6) show it is very likely that the conclusion always holds.

For $f(\theta) = \sin\theta + (\pi - \theta)\cos\theta$, its associated function $g$ that we have analyzed in Subsection 5.1 is actually an AD function by the following theorem, and the proof is presented in Appendix D.2.

**Theorem 5.6.** *For $f(\theta) = \sin\theta + (\pi - \theta)\cos\theta$, the associated function g is an AD function.*

Combining Proposition 5.4 (positive semi-definite), we have actually proved that part of determinant condition ($\det \boldsymbol{M} \geq 0$) holds theoretically.

Note that $g(x, y) \equiv 0$ is a trivial AD function, and its associated matrices are always $\boldsymbol{0}$. Obviously, this function is not powerful enough, so we introduce a special class of AD function which is more powerful.

**Definition 5.7.** *(PAD function) An AD function g is called a positive-definite angular distance (PAD) function if for any given angles $0 = \theta_0 < \theta_1 < \cdots < \theta_m < 2\pi$ with no $1 \leq i < j \leq m$ s.t. $\theta_i + \pi = \theta_j$, its associated matrix $\boldsymbol{M}$ is always positive-definite.*

PAD function does exist. For example, if $f(x) = \pi - x$, then $g$ is a PAD function by Theorem 3.1 in Du et al. (2019c).

If $f(\theta) = \sin\theta + (\pi - \theta)\cos\theta$, we have proved that its associated function $g$ is an AD function, and will show that $g$ is a PAD function if the following conjecture holds.

**Conjecture 5.8.** *(criterion of PAD) If an AD function g is the associated function of a strictly decreasing function f, and $\forall \alpha, \beta$ s.t. $\alpha, \beta, \alpha + \beta \in (0, \pi)$, it holds that $f(\alpha)f(\beta) > f(\alpha + \beta)f(0)$, then g is a PAD function.*

If Conjecture 5.8 (criterion of PAD) holds, then by Lemma 4.2 (decreasing lemma) and Lemma 4.3 (log-concave lemma), $g$ is a PAD function. Then by Definition 5.7 (PAD function) and Conjecture 5.5 (non-positive determinant), the determinant condition in Theorem 5.1 always holds.

Therefore, Theorem 5.1 can be rewritten as the following form.

**Theorem 5.9.** *If Conjecture 5.5 (non-positive determinant) and Conjecture 5.8 (criterion of PAD) holds, then equation system equation 2 has no non-trivial solution, i.e., the above neural network converge only if aligning.*

Although we leave some conjectures in this subsection unproved, AD function is worth extensive research, since the alignment phenomenon in slightly general case can be rigorously proved directly by the conjectures concerned with AD function, and we actually apply decreasing, log-concave lemma and definition of PAD function in the proof of two special cases in Section 4 respectively. Also, AD function appeared implicitly in the previous literature, which indicates that AD function can not only be applied to the theoretical analysis of alignment phenomenon, but also other behaviour of neural networks.

## 6 EXPERIMENTS

In this section, we show some experimental results to verify Conjecture 5.5 (non-positive determinant) numerically. Since it is impossible to test all the AD functions, we select the associated function $g$ of $f(\theta) = \sin(\theta) + (\pi - \theta)\cos(\theta)$ as a representative. Note that it is actually equivalent to verify determinant condition in Theorem 5.1.

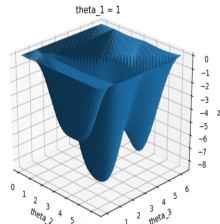 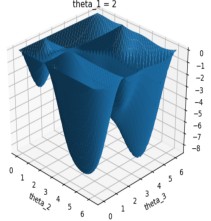 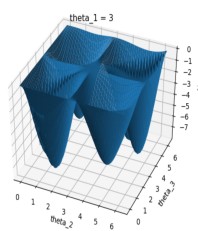

Figure 2: We fix $\theta_1 = 1, 2, 3$. $\theta_2$ and $\theta_3$ are variables. Also, $z = \min\{\det \boldsymbol{M}_1, \det \boldsymbol{M}_2, \det \boldsymbol{M}_3\}$.

First, we test the case when $m = 3$. Without loss of generality, we assume that $\theta_0 = 0$, then fix $\theta_1$ and rotate $\theta_2, \theta_3$ from 0 to $2\pi$. We choose $\theta_1$ to be 1, 2 and 3, and draw the minimum value of the

determinant of three associated matrix of $g$, which is defined in Definition 5.3 (associated matrix). The results are shown in Figure 2.

It is obvious that $z$ is always non-positive. Also, we test the case when $m = 4$. We fix $\theta_1$ and $\theta_4$. The results are similar to the case when $m = 3$ and shown in Figure 3:

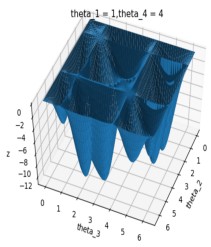 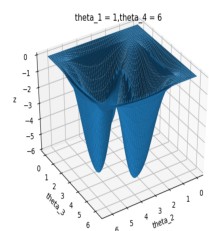 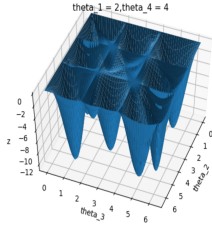

Figure 3: We fix $(\theta_1, \theta_4) = (1, 4), (1, 6), (2, 4)$. $z = \min\{\det \boldsymbol{M}_1, \det \boldsymbol{M}_2, \det \boldsymbol{M}_3 \ \det \boldsymbol{M}_4\}$.

For more general cases, it is difficult to draw figures due to the high dimension. Therefore, we randomly choose $\theta_1, \cdots, \theta_m$ and compute the minimum value of associated matrices $z$ for 100000 times for $m = 6$ and $m = 9$ respectively. Again, $z$ is always non-positive (see Figure 4).

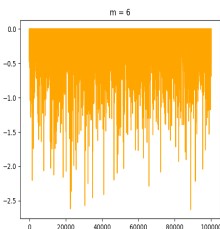 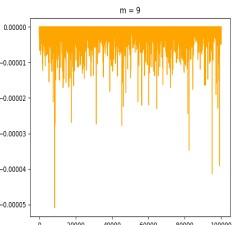

Figure 4: For $m = 6$ and $m = 9$, we randomly choose $\theta_1, \cdots, \theta_m$ and compute $z = \min\{\det \boldsymbol{M}_1, \cdots, \det \boldsymbol{M}_m\}$ for 100000 times respectively. For visualization, $z$ is clipped.

## 7 CONCLUSION

In this paper, we investigated the question that whether the elegant convergence theory for ultra-wide neural networks ($m = \Omega(N^6)$) could apply to practical neural networks ($m = O(N)$). We observed dissenting behaviors from the training in a simple two-layer teacher-student framework with ReLU activation and Gaussian inputs: The student neurons all get aligned with one of the teacher neurons, hinting that no local minima near any random initialization.

Following the empirical observation, we analyzed the local minima of the single-teacher multi-student setting. We proved that there is no local minima except students being aligned with the teacher for $m \leq 2$ and excluded the existence of local minima taking certain special forms for $m \geq 3$: (i) all students falling in a half plane, or (ii) all students with polar angles being some rational multiples of $\pi$.

Moreover, for general cases, we proposed a reduction from the existence of non-trivial local minima (i.e., local minima without alignment) to analyzing certain mathematical properties of a class of functions that we call angular distance (AD) functions. In particular, we showed strong empirical evidence that the such properties hold for the specific AD function, which in term serve as a sufficient condition for no existence of non-trivial local minima for general $m \geq 3$.

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
