# OpenReview forum: "Analysis of Alignment Phenomenon in Simple Teacher-student Networks with Finite Width"
_ICLR.cc/2021/Conference — Reject_

### Official Review · AnonReviewer1 · 2020-10-26

**Rating:** 3
**Confidence:** 4

**Review:**

[Summary]

This paper studies the optimization landscape of one-hidden-layer neural networks in the teacher-student setting, where the ground truth teacher network is one relu and the learner network is m>=2 relus. The paper proves that (1) when m=2, any stationary point (of the student network) has to be aligned with the teacher; (2) when m>=3, stationary points satisfying certain additional conditions have to be aligned with the teacher. Above, alignment means that the student network is perfectly equal to the teacher.

[Pros]

The one-hidden-layer teacher-student setting studied in this paper is an important and difficult theoretical problem. The reason why GD on an over-parameterized student could perfectly recover the teacher has been a somewhat long-standing open problem. This paper makes progress on this problem by proving the alignment property at m=2, and showing that the “determinant condition” implies alignment at m>=3.


[Cons]

I feel like the direction of studying alignment in this specific one-neuron teacher model may be a bit problematic, and may not be that significant or generalizable.

---First, results in this paper seem to be about the *stationary points*, not the *local minima* (which at least additionally requires the Hessian to be PSD). This can be seen from Eq (1) and (2) which is only the stationarity condition (gradient equals zero) and does not consider the Hessian. This means that all the subsequent claims are about stationary points and must also contain the saddle points. It seems like the authors may be unaware of this. (The paper talks about “local minima” where it really means gradient equals zero.)

---Second, the fact that the theory contains saddle points makes it somewhat confusing --- If a saddle point does exist, then it has to be non-aligned (because a saddle point must have non-zero loss). So the present result seems to imply that there is no saddle point at m=2 and (likely) no saddle point at m>=3, when the teacher is one relu. (Could the authors confirm or comment on this?)

---Further, even if we restrict attention to local minima, (Safran and Shamir 2018) already showed that when the teacher contains multiple neurons, the student network could have spurious local minima with non-zero risk (i.e. non-aligned minima). Therefore we could not hope for alignment for all local minima when the teacher has multiple neurons, which implies that the studying the alignment in the one-neuron teacher model may not be generalizable.

------
Thank the authors for the responses. While I agree with some of the points made by the authors (e.g. understanding the base case is interesting), I am still concerned about the significance of the result. Therefore I would like to keep my original evaluation.

---

> ### Author Response · Authors · 2020-11-22
> **Response to Reviewer 1**
>
> We thank the reviewer for the insightful and helpful comments.
>
> “Results in this paper seem to be about the stationary points, not the local minima.”
> We study stationary points and our conclusions apply to local minima. We established the central equation system (eq (2)) based on the first order conditions (eq (1)), as the main focus of this paper is about converged states rather than local minima. In principle, just as the reviewer said, the solutions to eq (2) may also contain saddle points. However, as we proved in the paper (or conjectured for the general cases), all solutions to eq (2) must be trivial, namely *all stationary points should be global minima*. Therefore, the saddle point issue that the reviewer worried about should never happen.
>
> “So the present result seems to imply that there is no saddle point at m=2 and (likely) no saddle point at m>=3, when the teacher is one relu.”
> Yes, our theorems (conjectures) imply that there is (likely) no saddle point.
>
> “We could not hope for alignment for all local minima when the teacher has multiple neurons, which implies that studying the alignment in the one-neuron teacher model may not be generalizable.”
> We didn’t hope for the alignment to generalize to all neural networks. While combining our results and Safran and Shamir (2018), we know that the alignment guarantee will break for networks with more than X teacher neurals, where X no more than 6 and likely no less than 2.
> In particular, having a clear understanding of the alignment in the basic case will help us understand why it does not generalize for more complicated cases.

---

### Official Review · AnonReviewer4 · 2020-10-28
**Reviews**

**Rating:** 5
**Confidence:** 3

**Review:**

This paper considers the problem of learning neural networks with finite width. More specifically, the authors consider the simple teach-student setting and show that the input weights of student neurons will align with the teacher neuron. The problem considered in this paper is interesting. However, the results derived in this paper only consider some simple setting which may limit its impact. Here are some concerns I have for the current paper:
1. The authors only consider the setting that the teach network only has one neuron. Thus it may not be reasonable to claim that the student neurons will align with one of the teacher’s neurons. If the authors can provide results for the teacher network with multiple neurons, the contribution of the current paper would be stronger.
2. Why the results only consider the case when d=2?
3. The presentation of the current paper needs to be improved. For example, for the proof sketches of Theorems 4.1, 4.4, it is unclear the roles of Lemmas 4.3, 4.5.
4. There is a recent work [1] also studying the learning of a single neuron model, the authors may want to compare with their results.
5. The introduction and related work sections are clear and thorough. However, there are some relevant papers [2,3] also study the teacher network setting, the authors may want to discuss them in these sections.

References:
[1] Frei, Spencer, Yuan Cao, and Quanquan Gu. "Agnostic Learning of a Single Neuron with Gradient Descent." Advances in Neural Information Processing Systems (NeurIPS), 2020

[2] Fu, Haoyu, Yuejie Chi, and Yingbin Liang. "Guaranteed recovery of one-hidden-layer neural networks via cross entropy." IEEE Transactions on Signal Processing 68 (2020): 3225-3235.

[3] Zhang, Xiao, et al. "Learning one-hidden-layer relu networks via gradient descent." The 22nd International Conference on Artificial Intelligence and Statistics. PMLR, 2019.

---

> ### Author Response · Authors · 2020-11-22
> **Respond to Reviewer 4**
>
> We thank the reviewer for the insightful and helpful comments.
>
> “Why the results only consider the case when d=2?”
> The problem is already very challenging even for the simplest d = 2 and m = 3 case. We would love to generalize the results for d > 2 in the future.
>
> “The presentation of the current paper needs to be improved.”
> Thanks for your suggestion, we will improve our paper accordingly.
>
> Thanks for the suggested references. We will cite them.

---

### Official Review · AnonReviewer2 · 2020-10-28
**Novel results on alignment in neural networks, but the current scope of the work is too narrow.**

**Rating:** 4
**Confidence:** 4

**Review:**

### Summary

This work considers learning one-hidden layer convex neural networks where the teacher is a single ReLU and the data follows a two-dimensional Gaussian distribution. The authors study the loss landscape of the population loss and provide conditions on critical points which imply that the neurons are aligned with the teacher network (I will call this an aligned solution). They show that for m<=2, any critical point is necessarily an aligned solution. For m > 2, they show that a critical point is an aligned solution in certain special cases (e.g., when all polar angles are rational multiples of pi). They further show that in the general case, under certain conjectures, a critical point is an aligned solution. For the latter result they introduce a novel concept of Angular Distance (AD) functions.

### Reason for score

Analyzing the non-convex loss landscape of neural networks is extremely challenging (even in seemingly "simple" cases) and I appreciate the effort of the authors to tackle it. While the theoretical results are novel and may be useful for analyzing more advanced settings, I think that the scope of the current manuscript is too narrow and lacks several important empirical results and discussions. Therefore, I don't think that the current version is ready for publication and I highly encourage the authors to improve the current paper (see my suggestions for improvement below).

### Pros
1.	The authors tackle a very challenging problem and provide novel theoretical results.
2.	The concept of AD functions is interesting and might be useful in other analyses of neural networks.

### Cons
1.	In this paper the authors only show results for d=2. This is highly restrictive. I understand that for d>2 the analysis might be intractable. Therefore, I think that the authors should add extensive experiments which show that alignment occurs in higher dimensions. To further strengthen the results, it would be interesting to see if some of the theoretical insights of d=2 can be also applied to higher dimensions (e.g., the analysis of AD functions). The authors mention d=2 only once in the paper. This should be clearly stated in the Abstract, Introduction and Conclusion sections.
2.	The authors cite the work of Tian (2020) but do not provide a clear comparison to this work. Specifically, it is not exactly clear what is the novelty of this work compared to Tian (2020) since the latter also shows that alignment occurs for any dimension.
3.	In the experiments section the authors do not empirically verify Conjecture 5.5. Therefore, it is not clear whether this is a realistic conjecture.
4.	The authors show two special cases where any critical point is an aligned solution. The key question is whether gradient descent converges to solutions which satisfy the conditions of these cases. This is not discussed and I encourage the authors to perform experiments to check if this holds.
5.	The authors should cite more works on student specialization, e.g., "Dynamics of stochastic gradient descent for two-layer neural networks in the teacher student setup" of Goldt et al. (2019) and "Dynamics of on-line gradient descent learning for multilayer neural networks" of Saad et al. (1996).

---

> ### Author Response · Authors · 2020-11-22
> **Response to Reviewer 2**
>
> We thank the reviewer for the insightful and helpful comments.
>
> “The authors mention d=2 only once in the paper. This should be clearly stated in the Abstract, Introduction and Conclusion sections.”
> We will emphasize that and consider adding more experiments for general settings in the future.
>
> “Comparison with Tian (2020)”
> Tian (2020) required a quite strong assumption that the gradient is zero for all possible inputs. Instead, our result is built on the most standard condition that gradient descent is stable (expected gradient is zero). We will add the discussion to the paper.
>
> “Empirically justify Conjecture 5.5?”
> Thanks for your suggestion. We will perform more experiments to verify the conjecture.
>
> “Perform experiments to verify whether it will converge”
> We agree that this is good to have in the paper, yet this is not the main focus. We will try to incorporate more experiments in the future.
>
> We will add the missing citations, and thank you for the references.

---

> > ### Comment · AnonReviewer2 · 2020-11-22
> > **Thanks for the clarifications**
> >
> > I read the other reviews and responses and did not change the score.

---

### Official Review · AnonReviewer3 · 2020-11-02
**Interesting ideas for characterizing the critical points of over-parametrized student network for a teacher network with one neuron**

**Rating:** 4
**Confidence:** 4

**Review:**

Summary of review:

This paper studies the optimization landscape of over-parametrized two-layer ReLU neural networks under an (isotropic) Gaussian input distribution. The main goal is to characterize the critical points of the over-parametrized student network, and show that these critical points "align" with the neurons of the teacher network. This is a well-motivated study since the alignment phenomenon is intriguing (Tian, 2020) but lacks a theoretical analysis.

Setting:

This paper focuses on a teacher network with a single two-dimensional neuron followed by a ReLU activation function. The student network contains multiple (two-dimensional) neurons followed by a ReLU activation function.

Results:

This paper presents a suite of results for the above setting, including:

(i) When the student network has two neurons, this paper shows that both neurons "align" perfectly with the neuron of the teacher network. Furthermore, the sum of the length of the two neurons is equal to the length of the neuron of the teacher network.

(ii) When the student network has more than two neurons, this paper shows that the above "alignment" result holds for two special cases:

-- (ii-a) when all the neurons of the student network are in the upper half-plane;

-- (ii-b) when all the polar angles are rational multiples of $\pi$. Furthermore, the polar angles satisfy a certain positive-definiteness condition in this special case.

-- For the general case, this paper discovers an interesting determinant condition that guarantees the condition of (ii-b), hence the "alignment" result. In general, it is not clear whether the determinant condition always holds, and this paper provides several conjectures/ideas that provide some preliminary evidence for why this condition should hold.

Limitation:

While the questions are well-motivated and the settings are quite standard, I am concerned that the techniques of this paper are too restrictive to the assumptions. In particular, these include:

(a) The dimension of the neurons being only two. This looks like a strong assumption, but is not mentioned or even discussed anywhere in the abstract and the introduction.

(b) The results for the student network having more than two neurons are still rather preliminary. While this paper does provide several interesting ideas that might help in ultimately resolving the question, as the results stand currently, it is quite unclear to the reviewer whether the conjecture(s) will hold or not.

Questions:

(a) The proof of Theorem 3.1 relies on that the critical point condition holds exactly. What if the critical point condition only holds approximately up to a small error of $\epsilon$? Would your analysis still extend there? This setting is more interesting because in general, it is difficult for gradient descent to converge to an exact critical point. Furthermore, this setting might only extend to allow for sampling errors in an empirical loss.

(b) What about the proof of Lemma 4.5 and Theorem 5.1? Would the analysis extend to allow for sampling errors or not?

Other comments:

- At the bottom of page 5, it will help to move the $\rho_0$ and $\rho_j$ outside of $\cos$. Right now, this equation looks a bit confusing..

---

> ### Author Response · Authors · 2020-11-22
> **Response to Reviewer 3**
>
> We thank the reviewer for the insightful and helpful comments.
>
> “What if the critical point condition only holds approximately up to a small error of ϵ? … What about the proof of Lemma 4.5 and Theorem 5.1? Would the analysis extend to allow for sampling errors or not?”
> This is a very good question for future work. We are also interested in answering these questions, yet these might not be the main focus of this paper.

---

### Decision · Program_Chairs · 2021-01-07
**Final Decision**

**Decision:**

Reject

**Comment:**

The reviewer seems to reach a consensus that the paper is not ready for publication at ICLR. One of the major issues seems to be that the paper only analyzes the case of $d=2$. (In the AC's opinion,  $d>2$ might be fundamentally more difficult to analyze than $d=2$).